# Understanding the Impact of Salt Stress on Plant Pathogens Through Phenotypic and Transcriptomic Analysis

**DOI:** 10.3390/plants14010097

**Published:** 2025-01-01

**Authors:** Hyejung Jung, Gil Han, Duyoung Lee, Hyun-Kyoung Jung, Young-Sam Kim, Hee Jeong Kong, Young-Ok Kim, Young-Su Seo, Jungwook Park

**Affiliations:** 1Department of Integrated Biological Science, Pusan National University, Busan 46241, Republic of Korea; 2Biotechnology Research Division, National Institute of Fisheries Science, Busan 46083, Republic of Korea

**Keywords:** climate change, comparative transcriptomic analysis, high salinity, plant pathogen, salt stress, salt tolerance, virulence

## Abstract

For plant diseases to become established, plant pathogens require not only virulence factors and susceptible hosts, but also optimal environmental conditions. The accumulation of high soil salinity can have serious impacts on agro-biological ecosystems. However, the interactions between plant pathogens and salinity have not been fully characterized. This study investigated the effects of salt stress on representative plant pathogens, such as *Burkholderia gladioli*, *Burkholderia glumae*, *Pectobacterium carotovorum* subsp. *carotovorum* (*Pcc*), *Ralstonia solanacearum*, and *Xanthomonas oryzae* pv. *oryzae*. Phenotypic assays revealed that *B. gladioli* and *R. solanacearum* are highly sensitive to salt stress, exhibiting significant reductions in growth, motility, and enzyme production, whereas *Pcc* showed notable tolerance. Pan-genome-based comparative transcriptomics identified co-downregulated patterns in *B. gladioli* and *R. solanacearum* under stress conditions, indicating the suppression of bacterial chemotaxis and type III secretion systems. Uniquely upregulated patterns in *Pcc* were associated with enhanced survival under high salinity, such as protein quality control, osmotic equilibrium, and iron acquisition. Additionally, the application of salt stress combined with the beneficial bacterium *Chryseobacterium salivictor* significantly reduced tomato wilt caused by *R. solanacearum*, suggesting a potential management strategy. This study underscores practical implications for effectively understanding and controlling plant pathogens under future climate changes involving salt stress.

## 1. Introduction

Plant diseases inflict substantial annual losses on a global scale, significantly affecting crop yields and food quality. The direct production damage to major agricultural crops caused by plant diseases is estimated to range from 20 to 40% [1,2]. The genera *Ralstonia*, *Xanthomonas*, *Pectobacterium*, *Pseudomonas*, *Burkholderia*, and *Dickeya* are the most well-known plant pathogens, with significant economic impacts on agriculture [3,4]. For plant diseases to become established, these pathogens depend not only on virulence mechanisms within the host organism, but also on environmental factors. Even susceptible plants exposed to destructive pathogens will not develop disease unless the environmental conditions are favorable for the plant pathogen. This “disease triangle” concept in plant pathology depicts the sophisticated relationships between a plant pathogen, a susceptible host, and environmental factors [5]. Plant pathogens have evolved to monitor and exploit favorable conditions for disease establishment in dynamic environmental conditions [6,7,8].

Over the past few decades, climate change has hindered the establishment of favorable conditions for biological members in agricultural environments [9]. The 2000s were 1.5 °C warmer than the early 1900s, and in particular, land-based warming has outpaced the increase in global average temperature [10]. The threat of climate change is not limited to changes in mean temperature, leasing to unanticipated consequences. For example, sea-level rise resulting from the melting of ice sheets causes salinization of freshwater and loss of agricultural land [10]. Climate change also accelerates the rate of soil evaporation, resulting in increased desertification and the accumulation of toxic substances [11]. Thus, agricultural environments consisting of plants and plant-associated microorganisms face various types of abiotic stresses.

Among these abiotic stresses, high salinity is a critical factor that severely affects the structure and composition of plant-associated bacteria [12,13,14]. Salt stress induces hyperosmotic and hyperionic states that inhibit bacterial growth and reduce bacterial biomass [15]. Disturbances in biological activities, including those associated with organic matter degradation, nutrient cycling, and soil enzyme activity, can occur when plant-associated bacteria are exposed to salt stress [16,17,18,19]. Plant-associated bacteria affected by salt stress include not only indigenous and commensal bacteria but also plant pathogenic bacteria. In general, the bacterial mechanisms to tolerate salt stress require enormous amounts of energy, making it difficult for plant pathogens to maintain full virulence in high salinity. In *Pseudomonas syringae* pv. *tomato* DC3000, a known causal agent of bacterial speck disease in tomato plants, growth reduction of up to 40% was observed under salt stress conditions [20]. Despite the impact of salt stress, the physiological changes in major plant pathogens under high salinity conditions remain unclear. Considering future climate changes, it is essential to identify the systemic mechanisms that respond to salt stress at the genomic level to fully control plant pathogens.

Transcriptome-based approaches are effective for studying direct responses to stressful conditions. Bacterial transcriptomes provide a quantitative and unbiased method to decipher primary changes in gene expression and provide a snapshot of the physiological state [21]. Liao et al. evaluated the genome-wide transcriptional profile of *Xanthomonas citri* pv. *citri* in response to cold stress using RNA-seq technology [22]. Global transcriptomic analyses of *P. syringae* have demonstrated osmotic stress-dependent transcriptional changes in the type III secretion system (T3SS) and type VI secretion system, which are associated with virulence factors [23]. Furthermore, comparative strategies for transcriptome data have been used to identify key factors that are either global or specialized across multiple species beyond the level of a single organism [24,25].

The objective of this study was to elucidate the diverse interactions between plant pathogens and salt stress. We hypothesized that salt stress would induce significant changes in gene expression patterns across plant pathogens, affecting their virulence and physiological mechanisms. Representative plant pathogens were carefully selected and analyzed for their growth and virulence-related functions under salt stress conditions. Cross-species comparative transcriptome analysis revealed key mechanisms of plant pathogens for adaptation to salt stress. Additionally, we assessed the potential of salt stress as a method for controlling the plant pathogen through the virulence assay with tomato hosts. This study provides in-depth insights into the response mechanisms of plant pathogens under high salinity and guides strategic approaches for managing plant diseases in agricultural environments increasingly affected by climate change.

## 2. Results and Discussion

### 2.1. Salt Stress Tolerance of Representative Plant Pathogens

Salt stress is one of the most harmful abiotic stresses caused by various factors, including climate change and human activity. Natural processes of soil salinization are primarily caused by rising sea levels and increased evaporation rates due to global warming. Anthropogenic factors, particularly agriculture—specifically irrigated crop systems—further accelerate this process [26]. Salt stress has emerged as a critical challenge for agricultural environments, with approximately 20% of cultivated land and 33% of irrigated agricultural land worldwide currently affected. High soil salinity, which leads to the production of reactive oxygen species (ROS) and ionic imbalance, is damaging to all organisms, from microorganisms to plants [27]. These environments can also alter the structure and composition of agro-biological ecosystems and affect their interspecies interactions. Thus, considering the “disease triangle” concept in plant pathology, high soil salinity is a formidable challenge for plant pathogens to overcome and adapt to.

To explore changes in the physiological activity of plant pathogens in response to salt stress, we selected and collected five representative plant pathogens through extensive literature mining (Table 1): *Burkholderia gladioli* BSR3, *Burkholderia glumae* BGR1, *Pectobacterium carotovorum* subsp. *carotovorum* (*Pcc*) PCC21, *Ralstonia solanacearum* GMI1000, and *Xanthomonas oryzae* pv. *oryzae* (*Xoo*) PXO99^A^. *R. solanacearum* ranks second among the top ten bacterial plant pathogens [4]. It is the causative agent of bacterial wilt and Moko disease in >200 plant species, including tomatoes, potatoes, shrubs, and trees [28]. Plant diseases caused by *R. solanacearum* result in an estimated USD 1 billion in annual losses, with severe effects in numerous developing countries [29]. *Xoo*, the causative agent of bacterial leaf blight disease, and *Pcc*, the causative agent of soft rot disease, were ranked fourth and tenth, respectively, in the top ten lists [4]. In addition to their economic impact on agriculture, the virulence mechanisms of these two plant pathogens are important models for plant pathology [4,30]. *B. glumae* and *B. gladioli* have become major threats to the stability of rice production in several countries [31,32]. These two pathogens cause similar symptoms such as bacterial panicle blight and seedling rot and are often isolated from the same diseased plants [33]. Current global climate change may lead to outbreaks of plant pathogenic *Burkholderia* species owing to their wide range of optimal temperatures and ecological niches [3,34].

We investigated the effects of salt stress on bacterial growth. A significant reduction in growth and development under salt stress, specifically at 200 mM NaCl, has been observed in various plant species, including tomato, wheat, and water dropwort [40,41,42]. In addition, salt tolerance in microorganisms is commonly evaluated using 200 mM NaCl in many studies, as this concentration effectively induces salt stress, providing a suitable challenge for assessing microbial growth and adaptation [43,44,45]. Therefore, we selected 200 mM NaCl as the standard concentration for all experiments to ensure a consistent and reliable assessment of plant pathogens’ responses to salt stress. Under salt stress conditions, different growth responses were observed among the five plant pathogens (Figure 1). The growth rate of *R. solanacearum* decreased most significantly under salt stress conditions (*p*-value < 0.001). While *R. solanacearum* reached a stationary phase with an optical density at 600 nm (OD_600_) of 3.3 after 36 h under normal conditions, it did not reach an OD_600_ of 1.0 under salt stress conditions. The growth curve of *B. gladioli* also showed a significant difference compared with that of the control samples (*p*-value < 0.001). When the stationary phase was reached under salt stress conditions, its OD_600_ level was approximately 65.4% of that under normal conditions. Salt-treated cultures of *Xoo* exhibited a slower growth rate (*p*-value < 0.05) than the control samples throughout the analysis period.

In contrast, there were no significant differences in the growth responses of *B. glumae* and *Pcc* under salt stress conditions (Figure 1). Although *B. glumae* showed a slower growth rate in the presence of salt stress for 6 to 12 h, the growth curve continuously increased to match normal conditions up to 36 h. *B. glumae* and *B. gladioli* share several features including the same rice host, disease symptoms, and virulence mechanisms involving toxoflavin production [33,46]. Despite these physiological similarities, differences in salt stress tolerance may arise from differences in transcriptional responses at the genomic level. Interestingly, salt stress had no effect on the growth of *Pcc*, indicating that it is a slight halophile capable of surviving in 1–3% salinity. To successfully establish plant diseases, pathogens must first adapt to the environment on the attachment surface and grow sufficiently to initiate an infection [47]. These results confirm that, among the five plant pathogens, *Pcc* is in an advantageous position to colonize host plants in soil environments with high salinity.

### 2.2. Effects of Salt Stress on Bacterial Motility

Considering the importance of bacterial motility in plant pathology, we investigated the effects of salt stress on the swimming motility of five plant pathogens using semi-solid agar plates. Under normal conditions, most plant pathogens displayed active motility, with robust swim halos (Figure 2). Notably, *R. solanacearum* and *Pcc* exhibited the largest swim halos, at 6.16 ± 0.23 and 5.88 ± 0.48 cm, respectively. *B. glumae* and *B. gladioli* formed swim halos of 3.54 ± 0.59 and 3.24 ± 0.37 cm, while *Xoo*, though less motile than other pathogens, formed a halo with a diameter of 1.79 ± 0.32 cm.

However, under salt stress conditions, most plant pathogens except for *Pcc* lost almost all motility (Figure 2). *Pcc* demonstrated relative tolerance to salt stress in terms of motility, exhibiting a swim halo of 3.31 ± 0.15 cm under 200 mM NaCl. This is consistent with the results of bacterial growth assays (Figure 1), indicating that *Pcc* retains some motility despite salt stress. In contrast, *R. solanacearum*, which showed the most active motility under normal conditions, completely lost motility in the presence of 200 mM NaCl, with no detectable swim halos (*p*-value < 0.0001). Similar results were observed for *B. gladioli* and *B. glumae*, with swim halos reduced to 0.85 ± 0.12 and 0.90 ± 0.18 cm, respectively. Salt stress inhibited the motility of these two pathogens by approximately 75% compared with normal conditions (*p*-value < 0.001).

Motility with a flagellar structure is considered an important virulence factor and is pivotal for plant–pathogen interactions, as it helps pathogens invade plant tissues [48]. Mutations in the flagellar genes of motile plant pathogens significantly decrease their virulence in host plants [49,50]. It has also been reported that the expression of flagellar biosynthesis proteins among representative plant pathogens is commonly increased in a host environment-dependent manner [25]. For *R. solanacearum*, motility is generally essential during the early stages of infection, enabling the pathogen to invade plant hosts through root wounds, colonize xylem vessels, and spread rapidly up the stem to reach optimal bacterial densities and achieve full virulence. Tans-Kersten et al. demonstrated that the biological function of the flagellum is essential for virulence in *R. solanacearum* in the early stages of disease development before it reaches the stem [50]. Additionally, the association between flagellar-dependent motility and bacterial virulence has been highlighted in *Burkholderia* species [51,52]. In this context, our results suggest that high soil salinity is a critical obstacle to host infection by plant pathogens, potentially hindering their ability to invade host plants.

### 2.3. Effects of Salt Stress on Extracellular Enzyme Production

Under salt stress conditions, we evaluated the production of cell wall-degrading enzymes as another major virulence factor in plant pathogens. Protease- and cellulase-producing plant pathogens were effectively identified using skim milk agar and carboxymethyl cellulose (CMC) agar plates, respectively. The assay results on skim milk agar plates revealed that *B. gladioli*, *B. glumae*, *Pcc*, and *R. solanacearum* exhibited high proteolytic activity, with halo diameters ranging from 2.42 to 4.80 cm (Figure 3a). Among these, *Pcc* exhibited the highest activity, with a halo diameter of 4.80 ± 0.48 cm, while *B. glumae* showed the weakest activity, with a halo diameter of 2.42 ± 0.37 cm. Under salt stress conditions, *B. glumae* and *R. solanacearum* completely lost their proteolytic activity, showing a 100% reduction in halo diameter (*p*-value < 0.0001). Moreover, proteolytic activity was significantly reduced in *B. gladioli* and *Pcc*, showing >40% inhibition compared to their activity under normal conditions. The halo diameters observed in salt-supplemented plates were 2.28 ± 0.31 cm for *B. gladioli* and 2.75 ± 0.41 cm for *Pcc*, with *B. gladioli* exhibiting lower proteolytic activity compared to *Pcc*.

Interestingly, the halo diameters on CMC agar plates remained consistent on both normal and salt-supplemented plates. Only three plant pathogens—*Pcc*, *R. solanacearum*, and *Xoo*—produced clear zones of cellulolytic activity (Figure 3b). These three pathogens were observed to have similar halo diameters in salt-supplemented plates (1.93–2.20 cm) compared to normal plates (1.97–2.26 cm). These results indicate that salt stress has little effect on the cellulase production capacity of plant pathogens.

When plant pathogens reach an optimal bacterial density in plant tissues, they begin to produce a variety of cell wall-degrading enzymes, such as proteases, cellulases, pectinases, and xylanases [53]. These enzymes destroy plant tissues, allowing plant pathogens to invade effectively [54]. The importance of cell wall-degrading enzymes in plant pathology has been demonstrated by genome analyses, which have shown that 40% of plant pathogens encode these genes [55]. We noted a slight inhibition of proteolytic activity in *Pcc* under salt stress conditions, although this was not a complete loss as observed in *B. glumae* and *R. solanacearum* (Figure 3a). Several studies have revealed that the virulence mechanisms of the genus *Pectobacterium* primarily depend on the production and secretion of large amounts of plant cell wall-degrading enzymes such as pectate lyase and protease [56,57,58]. A mutation in the transcriptional regulator *rpfA*, which is involved in the production of extracellular enzymes, significantly reduced the virulence of *Pcc* [59]. Moreover, proteases play regulatory roles, enabling appropriate responses to environmental changes and functioning as components of the protein quality control system to maintain cellular proteostasis [60,61]. Therefore, our results imply that *Pcc* does not fully escape the devastating effects of non-specific stress caused by high salinity, even though it shows a strong tolerance to bacterial growth and motility.

### 2.4. Construction of Pan-Genome Map for Plant Pathogens

Salt stress resulted in different responses of five plant pathogens in the phenotypic assays of bacterial growth, motility, and extracellular enzyme production (Figure 1, Figure 2 and Figure 3). To interpret the dynamic interactions between plant pathogens and salt stress, we considered *B. gladioli* and *R. solanacearum* as salt-sensitive plant pathogens and *Pcc* as a salt-tolerant plant pathogen and performed cross-species comparative transcriptomic analyses. A comparison of gene expression among these plant pathogens can provide deep insights into the key factors driving physiological changes.

Comparative transcriptomic analyses begin with the identification of orthologous gene clusters that are assumed to maintain their original functions [62]. This approach is useful, because orthologous and non-orthologous gene clusters provide a genetic backbone for comparing conserved or specific expression patterns across bacterial species. Pan-genome analysis is an efficient framework for estimating and modelling genetic diversity within study groups [63]. The pan-genome of a bacterial species includes a core genome containing genes shared by all the species, a dispensable genome containing accessory genes that exist in two or more species, and a unique genome specific to a single species.

Completely sequenced genomes of five plant pathogens available at the National Center for Biotechnology Information (NCBI) genome database (ftp://ftp.ncbi.nlm.nih.gov/genomes/, accessed on 15 June 2023) were used in this study. The genomic information for each species is summarized in Table 2. *B. gladioli*, *B. glumae*, and *R. solanacearum* exhibited multiple chromosomal replicons, ranging from 5.81 to 9.05 Mb, whereas *Pcc* and *Xoo* contained only one chromosome with relatively smaller size (4.84–5.24 Mb). The number of genes (4315–8021) was positively correlated with genome size. The pan-genome map of the five plant pathogens revealed 10,046 gene clusters from 26,953 genes (Figure 4a and Appendix A). The core genome consisted of 920 gene clusters, accounting for 18.6 to 32.3% of each pathogen. Although this pan-genome had an open structure, the scale of the core genome was expected to converge to a constant value, given the slope of the exponential decay (Appendix A). The dispensable genome contained 3188 gene clusters, accounting for 31.7% of the total pan-genome map. From the core and dispensable genomes, we identified that the salt-sensitive plant pathogens *B. gladioli* and *R. solanacearum* shared 1246 orthologous clusters (Appendix A). In addition, the pan-genome map contained unique genomes that varied at 1073–1299 gene clusters. Among these, *Pcc* showed the most non-orthologous clusters (41.8% of the total gene clusters). This result suggests that *Pcc* has flexible genomic content, reflecting greater diversity in genetic functionality than other plant pathogens.

To assess the functional robustness of the genomic components of the pan-genome map, clusters of orthologous groups (COG) analysis was performed based on protein similarity. Of the genes comprising the pan-genome, 21,546 (79.9%) genes were annotated in the 25 COG categories; of these, 8228 genes were part of the core genome, and 10,205 and 3113 genes were part of the dispensable and unique genomes, respectively. Enrichment analysis for each COG category revealed distinct distributions between the core and dispensable/unique genomes (Figure 4b). The core genome was enriched in genes involved in the translation, ribosomal structure, and biogenesis (J, 9.1%) and transcription (K, 11.1%) categories (Appendix A). According to the Central Dogma, these COG categories are critical for the flow of genetic information in organisms. Indeed, the core genome performs basic housekeeping functions regardless of the evolutionary distance between bacterial species [63]. Dispensable and unique genomes affect the adaptation and survival of an organism within a niche [63]. As changes occur from a core genome to a unique genome, non-orthologous clusters are acquired by horizontal gene transfer or evolve due to mutations in existing genes. Our COG results also showed that the function unknown (S, 7.5%) and mobilome: prophages, transposons (X, 6.0%) categories were more abundant in dispensable/unique genomes. The mobilome represents the entire set of mobile genetic elements in the genome. The selective introduction of specific genes into certain species and environments may have accelerated the accumulation of unidentified proteins whose functions have not yet been defined.

### 2.5. Co-Downregulated Patterns in Salt-Sensitive Plant Pathogens

To profile the gene expression under salt stress conditions, we sequenced the transcriptomes of salt-sensitive (*B. gladioli* and *R. solanacearum*) and salt-tolerant (*Pcc*) plant pathogens. For each pathogen, RNA-seq libraries were constructed for both control (0 mM NaCl) and treatment (200 mM NaCl) conditions, with three biological replicates per condition. Transcriptome sequencing yielded an average of 45,447,416, 38,559,098, and 66,648,169 paired end reads for *B. gladioli* and *R. solanacearum* and *Pcc*, respectively (Appendix A). All of them exhibited high-quality sequencing reads with an average Phred score of 36 (Appendix A). After quality filtering and coordinate pairing, the number of clean reads per RNA-seq library ranged from 32,858,742 to 84,151,262 (Appendix A). The clean reads were then mapped against reference genomes that were previously used for pan-genome construction. The average mapping percentage of salt-sensitive *B. gladioli* and *R. solanacearum* was 88.7% (27,304,434–46,600,420 reads), whereas that of salt-tolerant *Pcc* was 91.1% (53,117,326–77,994,015 reads). The relative expression levels of 16,752 genes were quantified by readcount and RPKM (reads per kilobase of transcript per million mapped reads) values for downstream analyses. The raw sequencing data reported in this study were deposited in the NCBI Gene Expression Omnibus (GEO) database (https://www.ncbi.nlm.nih.gov/geo/, 24 July 2024) under accession number GSE272984.

In terms of the effects of salt stress on salt-sensitive plant pathogens, we focused on co-downregulated patterns, because salt stress causes diverse non-specific damage to organisms. The commonality in expression patterns among the study groups indicates conservation or constraints in the regulatory mechanisms against external changes [64]. To this end, we first analyzed differentially expressed genes (DEGs) under salt stress conditions. Gene expression profiling of *B. gladioli* and *R. solanacearum* revealed a clear dichotomy in the principal dimension between salt stress and normal conditions (Appendix A). When applying the stringent criterion of fold change (salt stress/control) ≥ 2 and *p*-value < 0.05, 349 upregulated and 378 downregulated DEGs were identified in *B. gladioli* (Appendix A). *R. solanacearum* exhibited 791 DEGs under salt stress conditions, of which 274 and 517 were upregulated and downregulated, respectively (Appendix A). In particular, the high distribution of downregulated genes in *R. solanacearum* was consistent with the results of the phenotypic assays showing greater sensitivity to salt stress than other plant pathogens (Figure 1, Figure 2 and Figure 3).

All downregulated DEGs in salt-sensitive plant pathogens were divided into genomic components based on a pan-genome map (Figure 5a). The results showed that 639 downregulated DEGs were distributed in both core and dispensable genomes, of which 288 were from *B. gladioli* and 351 were from *R. solanacearum* (Figure 5a). The unique genomes contained 90 and 166 downregulated DEGs in *B. gladioli* and *R. solanacearum*, respectively. Furthermore, the alignment of overlapping downregulated DEGs revealed candidates with 369 orthologous gene clusters for expression comparison. Although the distribution of DEGs was evident under salt stress conditions, the corresponding clusters may not necessarily share the same direction of gene regulation in the comparative groups. We investigated whether there were downregulated genes between two plant pathogens constituting each cluster using heatmap analysis. We found that most of these (245 gene clusters) were downregulated in both plant pathogens, of which 39 gene clusters showed more than 2-fold downregulation under salt stress conditions (Figure 5b and Appendix A). In contrast, 124 gene clusters showed different expression patterns when upregulated in a single plant pathogen. In conclusion, 245 gene clusters were identified as being co-downregulated in salt-sensitive plant pathogens, which may represent functions suppressed by salt stress.

To understand the biological roles of the co-downregulated patterns, all DEGs from the 245 gene clusters were searched against the biological process domains in the GO database. Each GO term provides a systemic unit that comprehensively describes the properties of genes and their products in all organisms. GO enrichment analysis demonstrated that 203 downregulated DEGs were subjected to a total of 121 GO terms; among them, there were 13 terms with a *p*-value < 0.01 (Table 3). Most enriched terms with co-downregulated patterns suggest a variety of virulence mechanisms in plant pathogens. For example, nitrate assimilation (GO:0042128, *p*-value = 3.58 × 10^−8^) was a key factor promoting *R. solanacearum* virulence during the initial stage of infection, and mutations in the *nasA* gene related to nitrate assimilation resulted in reduced virulence and delayed stem colonization after soil inoculation of tomato hosts [65]. We also found enrichment of chemotaxis (GO:0006935, *p*-value = 5.51 × 10^−7^), signal transduction (GO:0007165, *p*-value = 1.27 × 10^−5^), and protein secretion (GO:0009306, *p*-value = 0.005) terms, which are essential for plant pathogen–host plant interactions [66,67,68,69]. Interestingly, Abdelwahed et al. demonstrated that when bacteria are exposed to abiotic stresses such as cold, heat, and ROS, multiple genes are downregulated to conserve biological energy [70]. These conserved energy and resources are then focused on fundamental survival rather than the growth or expansion of the organism through the stress response process. Based on our analysis and those of previous studies [70,71,72], we conclude that in harsh environments with salt stress, salt-sensitive plant pathogens are inevitably forced to change their virulence mechanisms to suboptimal strategies for survival.

Figure 5c summarizes the molecular network of bacterial chemotaxis (map02030) for how DEGs with co-downregulated patterns are involved in this system. A total of 15 DEGs were distributed across several important components: (1) chemoreceptors (methyl-accepting chemotaxis proteins, MCPs) (2) a histidine kinase, CheA; (3) a receptor-coupling protein, CheW; (4) receptor-modification enzymes, CheR, and CheB; (5) a response regulator, CheY; and (6) a regulator-modification enzyme, CheZ. Bacterial chemotaxis is a signal transduction process through which microorganisms migrate in response to environmental stimuli, including the ability to migrate to abundant nutrients [73]. It has also been reported that among representative plant pathogens, MCPs are in planta environment-dependently upregulated for movement within host tissues [25]. Severe abiotic stresses can cause drastic changes in these bacteria. As part of the stress response, bacteria suppress flagellar motility and simultaneously promote biofilm formation to protect themselves [74,75]. Therefore, the function of the bacterial chemotaxis system would be an unnecessary investment in the immotile situation of salt-sensitive plant pathogens due to salt stress.

Furthermore, multiple DEGs with co-downregulated patterns were associated with bacterial secretion (map03070), concentrated only in the T3SS (Figure 5d). The primary virulence determinant in most Gram-negative animal and plant pathogens is the presence of T3SS [69,76,77]. Effector proteins secreted by the T3SS subvert biological metabolism and suppress immune responses in host plants through mechanisms such as molecular mimicry [78,79,80]. However, considering that the operating principle of T3SS is to be directly injected into host cells rather than extracellularly, high T3SS expression may be a luxury for plant pathogens during salt stress-induced motility impairment. Therefore, the co-downregulated patterns of plant pathogens appear to be a response process to cope with salt stress, although it does not confer complete tolerance.

### 2.6. Uniquely Upregulated Patterns in Salt-Tolerant Plant Pathogens

The divergence in expression patterns across study groups provides crucial evidence for the existence of organism-specific functional responses [81]. Using the same criteria as mentioned above, 11 upregulated and 21 downregulated DEGs were identified in salt-tolerant *Pcc* (Figure 6a). We focused on DEGs that showed different expression patterns from those of salt-sensitive plant pathogens. A heatmap of the three plant pathogens showed similar expression patterns for the four DEGs (Figure 6b). In contrast, the genes PCC21_RS18685, PCC21_RS19340, and PCC21_RS20250 were upregulated only in *Pcc* and downregulated in *B. gladioli* and *R. solanacearum*. Including upregulated DEGs, PCC21_RS00435, PCC21_RS19335, PCC21_RS19345, and PCC21_RS21930 in the unique genome, we finally identified seven DEGs as uniquely upregulated patterns in *Pcc*.

Figure 6c shows a network structure of genes associated with the uniquely upregulated patterns, which was built from background information of protein–protein interaction (Appendix A). Genes PCC21_RS19335, PCC21_RS19340, and PCC21_RS19345 were interconnected with seven genes, and genes PCC21_RS20250 and PCC21_RS21930 each interacted with five or more genes. Clustering analysis revealed that these genes were divided into five clusters defined by functional systems, including iron ion transporter, Cpx stress response system, potassium transporter, fructose and mannose metabolism, and glyoxylate metabolism. Surprisingly, most of the functions associated with uniquely upregulated patterns may play important roles in the response strategies to salt stress.

One of the cytotoxic effects of salts is the induction of protein denaturation through the removal of water molecules from the protein surface [82]. Among the uniquely upregulated patterns, the cell envelope stress modulator CpxP (PCC21_RS20250) may be involved in the folding and degradation of abnormal proteins. As an adaptor protein, CpxP interacts with subsets of misfolded proteins and delivers them to DegP protease for degradation [83,84]. Isaac et al. demonstrated that mutation of the *cpxP* gene in *Escherichia coli* does not inhibit cytotoxicity by preventing degradation of misfolded P pilus proteins [83]. CpxP also acts as a negative regulator of the Cpx-envelope stress system, which responds to various signals such as salt, elevated pH, and hormones, by inactivating the sensor kinase CpxA [85,86]. Thus, the upregulation of CpxP may indicate not only direct protection against salt stress but also the maintenance of cellular balance from excessive stress response systems in salt-tolerant *Pcc*.

Another factor responsible for the response to salt stress in *Pcc* is the KdpF membrane protein (PCC21_RS21930), which is part of the K^+^-transporting P-type ATPase Kdp complex. At high external salt concentrations, K^+^ is a representative solute that achieves osmotic equilibrium in the cell [87]. A variety of bacterial species use the *kdp* operon for the uptake of osmoregulatory solute [88,89]. Of these, the KdpF protein is required to stabilize the Kdp complex, and inactivation of KdpF does not affect growth under low potassium conditions but significantly reduces the ATPase activity of the Kdp complex in vitro [90]. Interestingly, our analysis revealed that the *kdpAB* genes were upregulated in all three plant pathogens, whereas the *kdpF* gene was uniquely upregulated in *Pcc* (Figure 6b). These results support the key role of KdpF in discriminating between salt tolerance and sensitivity to plant pathogens. In addition, the uniquely upregulated patterns in *Pcc* involved the TonB-dependent siderophore receptor (PCC21_RS19340) and IucA/IucC family siderophore biosynthesis protein (PCC21_RS19345) for the efficient acquisition of iron ions. Iron is an essential nutrient for bacterial growth and is involved in various biological processes, including DNA synthesis, the tricarboxylic acid cycle, and biofilm formation [91]. However, iron availability is restricted under high soil salinity [92,93]. Thus, it appears that *Pcc* promotes the expression of siderophore-related genes in response to salt stress. In support of this notion, Hoffmann et al. reported that the growth of salt-stressed bacteria was significantly improved when the cells were supplied with excess iron [93].

### 2.7. Effect of Salt Stress on the Virulence of R. solanacearum

Among the five plant pathogens, *R. solanacearum* was the most sensitive to salt stress. Salt stress severely affected the growth of *R. solanacearum* compared to that in rich media and shut down key mechanisms associated with virulence (Figure 1, Figure 2 and Figure 3). Common downregulated genes also indicated a systematic mechanism supporting this sensitivity of *R. solanacearum* (Figure 5). These results led us to examine whether the salt treatment of tomato plants under soil conditions prevented bacterial wilting by inoculating *R. solanacearum*. Considering the non-specific damage caused by salt treatment, we used the beneficial bacterium *Chryseobacterium salivictor* NBC122, which has been reported to reduce salt stress in plants [94]. The genome of *C. salivictor* NBC122 contains various genes such as those involved in aromatic compound degradation and Por secretion systems, which are required to cope with complex environmental stresses associated with high salinity [95].

As shown in Figure 7, tomato seeds coated with *C. salivictor* were sown in normal and salt-supplemented soils, and 4–5-week-old plants were inoculated with *R. solanacearum* and scored for disease severity. Tomato plants grown in salt-supplemented soil showed normal growth and development, owing to the protective effect of *C. salivictor* against salt stress (Figure 7). *R. solanacearum*-inoculated plants in normal soil showed mild symptoms by day 5 and severe bacterial wilt by day 11 with a disease severity of 3.0 or higher. In contrast, salt stress significantly reduced the incidence of plant diseases caused by *R. solanacearum* (*p*-value < 0.001). Disease development in inoculated plants in salt-supplemented soil was delayed, with disease severity not reaching 1.0 until day 15. These results indicate that the combined application of salt stress and beneficial bacteria can effectively prevent and control salt-sensitive plant pathogens.

## 3. Materials and Methods

### 3.1. Plant Pathogens and Culture Conditions

The plant pathogens used in this study are listed in Table 1. Three plant pathogens—*B. gladioli* BSR3, *B. glumae* BGR1, and *Pcc* PCC21—are part of our laboratory’s collection. In contrast, *R. solanacearum* GMI1000 and Xoo PXO99^A^ were obtained from Dong-A University and Kyung Hee University in the Republic of Korea, respectively. *B. glumae* BGR1 was cultured in Luria-Bertani (LB) broth (BD Difco, Franklin Lakes, NJ, USA) at 37 °C. *B. gladioli* BSR3 and *Pcc* PCC21 were cultured in LB broth at 28 °C. *R. solanacearum* GMI1000 and *Xoo* PXO99^A^ were cultured at 28 °C in casamino acid peptone glucose (CPG, 1 g of casamino acids/L, 10 g of peptone/L, 5 g of glucose/L) and tryptic soy broth (BD Difco) media, respectively. All liquid cultures were incubated in a shaking incubator at 200 rpm.

### 3.2. Screening for Salt Stress Tolerance

For the salt stress tolerance analysis, plant pathogens were cultured in a medium containing 200 mM NaCl. Overnight bacterial cultures were adjusted to an OD_600_ of 0.01. The seed cultures were then inoculated into test tubes containing either normal medium (control) or NaCl-supplemented medium (salt stress). Samples were incubated at the optimal temperature for each pathogen for 48 h. Bacterial growth was monitored at 3 h intervals at 600 nm using a UV spectrophotometer (Shimadzu, Kyoto, Japan).

### 3.3. Motility Assay

To assess bacterial motility under salt stress conditions, a swimming medium was prepared using 0.3% agar plates containing 200 mM NaCl. An aliquot (approximately 3–5 μL) of bacterial cultures, adjusted to an OD_600_ of 1.0, was inoculated at the center of each plate. The inoculated plates were allowed to dry for 30 min before incubation for 24 h at the optimal temperature. The bacterial motility was measured as the distance from the point of inoculation to the periphery of the plate. Samples inoculated on agar plates without NaCl supplementation served as controls.

### 3.4. Screening for Extracellular Enzyme Production

Plant pathogens were assayed for production of virulence-related hydrolytic enzymes. Extracellular proteolytic and cellulolytic activities were assessed on 1% skim milk (BD Difco) and CMC (Junsei, Tokyo, Japan) agar plates, respectively, supplemented with 200 mM NaCl. A 4 mm sterile cork borer was used to create a central well in each plate, which was then inoculated with 20 μL of bacterial cultures adjusted to OD_600_ of 1.0. After incubating for 3 d at the optimal temperature, proteolytic activity was indicated by the formation of clear zones around the inoculated site. For the detection of cellulolytic activity, CMC agar plates were stained with 0.1% Congo red solution (Junsei) for 15 min, followed by washing twice with 1 M NaCl to observe clear zones.

### 3.5. Pan-Genome Analysis

A pan-genome analysis was performed using a pan-genome analysis pipeline (PGAP) tool [96]. The accession numbers of all genomes are listed in Table 2. The protein-coding sequences (CDSs) from all genomes were organized into gene clusters using the Gene Family method, involving comparison of each protein with all other proteins using basic local alignment search tool (BLAST) analysis. Genes with >30% identity, >30% coverage, and E-value of <10^−5^ were grouped into orthologous clusters using the Markov clustering algorithm [97]. A pan-genome map, including core, dispensable, and unique genomes, was constructed using an optimized algorithm incorporated into the PGAP tool, based on a presence/absence matrix.

Subsequently, each genomic repertoire of the pan-genome was classified according to COG functional categories [98]. Protein references with functional information were downloaded from COG database of NCBI (accessed in October 2023). Only alignments that met the criteria of >30% identity, >30% coverage, and an E-value of <10^−5^ were retained using the DIAMOND tool [99]. The distribution of genes in the 25 COG categories was compared among genomic repertoires.

### 3.6. RNA Extraction and Sequencing

Overnight cultures of salt-sensitive and salt-tolerant plant pathogens were inoculated at a 1:100 dilution in fresh medium and grown until the early exponential phase. The seed cultures were exposed to 200 mM NaCl for 2 h. Cultures without NaCl were used as controls. For sequencing library preparation, 4 mL of bacterial culture was harvested by centrifugation at 5000× *g* for 10 min and then stabilized using RNAprotect Bacteria Reagent (Qiagen, Valencia, CA, USA). Total RNA was extracted using the RNeasy Mini Kit (Qiagen) following the manufacturer’s protocol. RNA quality and quantity were measured using an Agilent 2100 Bioanalyzer (Agilent Technologies, Santa Clara, CA, USA). Subsequently, the ribosomal RNA was depleted using the NEBNext rRNA Depletion Kit (New England Biolabs, Ipswich, MA, USA), and the enriched RNA was used to construct cDNA libraries using the TruSeq Stranded Total RNA Library Prep Gold Kit (Illumina, San Diego, CA, USA), according to the manufacturer’s protocol. Paired-end next-generation sequencing was conducted using the Illumina NovaSeq 6000 platform (Macrogen, Seoul, Republic of Korea).

### 3.7. DEGs Analysis

The quality of raw reads was evaluated using the FastQC tool [100]. After trimming adapter sequences, reads with low quality (defined as having >50% of their length with a Phred score of ≤28) were filtered out using the FASTX-Toolkit version 0.0.13 [101]. Subsequently, an in-house Python script was employed to eliminate empty reads by comparing the forward and reverse coordinates. High-quality paired reads were aligned against the reference genomes (Table 2) using the Burrows-Wheeler Aligner tool [102]. Sequence alignment/map files were converted into binary alignment/map files using SAMtools version 1.13, which were then sorted and indexed [103]. To quantify expression levels, the FeatureCounts module in the Subread package was utilized to calculate the number of reads mapped to the CDSs [104]. Read counts for each gene were normalized using the RPKM method [105]. DEGs were identified by comparing gene expression levels between the two experimental groups using the edgeR package [106]. Genes showing differential expression with a greater than two-fold change and a *p*-value of <0.05 were considered as DEGs in this analysis.

### 3.8. Comparative Transcriptomic Analysis

For comparative transcriptomic analysis, the downregulated DEGs of salt-sensitive plant pathogens were assigned to orthologous gene clusters in the pan-genome map. Initially, gene clusters containing downregulated DEGs were selected. We then compared the expression levels of all genes within each cluster under salt stress and normal conditions, defining the co-downregulated patterns that were adversely affected by salt stress. In addition, the expression levels of DEGs in the salt-tolerant plant pathogen were directly compared with those in salt-sensitive plant pathogens. Genes that were upregulated only in the salt-tolerant plant pathogen or showed upregulation in unique gene clusters were defined as uniquely upregulated patterns.

To understand the biological systems associated with salt stress, functional enrichment analysis was conducted based on GO biological process terms [107]. The GO information on plant pathogens was extracted from the UniProt-GOA database [108]. The identified proteins were converted into UniProt entries and mapped to GO IDs. The statistical significance of the enriched systems was determined using a hypergeometric distribution test implemented with the phyper function in the R environment [109]. Statistical significance was set at a *p*-value of ≤0.05. The KEGG database was used for the metabolic pathway analysis [110]. The KEGG mapper tool was used to annotate genes associated with significantly enriched GO terms using KEGG orthologs [111]. Pathway networks based on molecular interactions and reactions were constructed using the Cytoscape version 3.10.1 [112].

Furthermore, the interaction network for uniquely upregulated patterns was explored using the STRING database [113]. The STRING database collates information from various sources, including experimental data, computational prediction methods, and data mining results. Uniquely upregulated genes in the salt-tolerant plant pathogen were retrieved from the *Pcc* reference, and unmatched genes were further retrieved from references of *Pectobacterium brasiliense* and *E. coli*, which are well-studied in genetic research. In this analysis, we considered interactions between proteins in terms of both functional and physical associations. The minimum score required for each interaction was set to a confidence threshold of 0.40 (medium level). To explore subnetworks from similar interacting genes and define their functions, Markov clustering analysis based on the stochastic process was applied with default parameters.

### 3.9. Virulence Assay Under Salt Stress Conditions

The influence of salt stress on the virulence of *R. solanacearum*, which was the most sensitive to salt stress among the five representative plant pathogens, was evaluated using a natural soil soaking method [25]. To mitigate plant damage caused by salt stress, we utilized *C. salivictor* NBC122, a beneficial microorganism reported to protect plants against salt stress [95]. Tomato seeds (cv. Zuiken) were surface-sterilized with 10% sodium hypochlorite and subsequently washed with distilled water. The sterilized seeds were agitated in bacterial cultures of *C. salivictor*, adjusted to an OD_600_ of 1.0, for 2 h. Thereafter, *C. salivictor*-treated seeds were sown at a depth of 3 cm in plastic pots (12 cm in diameter), each containing 1 L of salt-supplemented soil (100 mM NaCl). Plant growth conditions were as follows: day/night cycle of 12 h at 25 °C/12 h at 23 °C and a relative humidity of 50%.

For pathogen inoculation, *R. solanacearum* was cultured in CPG medium at 28 °C for 48 h, then centrifuged at 4000× *g* for 5 min and washed twice with distilled water. Each 4–5-week-old tomato plant was treated with a bacterial suspension at a final concentration of approximately 1 × 10^8^ colony forming unit (CFU)/g soil by pouring it into the soil near the plant roots. The uninoculated and inoculated plants were cultivated in a growth chamber with a 14/10 h day/night cycle at 28 °C. Disease severity in tomato plants was monitored after 15 days using the following scale: 0, no wilting; 1, 1–25% wilting; 2, 26–50% wilting; 3, 51–75% wilting; and 4, 76–100% wilting or death.

### 3.10. Statistics

Statistical analyses were performed using the R v3.6.3 environment. At least three biological replicates were used in each experiment. For the virulence assay, disease severity data from repeated experiments were pooled after confirming homogeneity of variance across at least 10 plants. Analysis of variance (ANOVA) was performed using the general linear model procedure, and means were separated using the least significant difference test at *p*-value < 0.05. The statistical details of the bioinformatics data are mentioned directly in the Section 3.

## 4. Conclusions

With climate change, high soil salinity is one of the most widespread abiotic stresses in agro-biological ecosystems, adversely affecting plants, indigenous microorganisms, and plant pathogens. We need to view plant pathogens and abiotic stress from the perspective of their complicated interactions, as in the concept of the “disease triangle”. The differential activity of plant pathogens in growth, motility, and extracellular enzyme production suggests that their virulence may significantly differ in high soil salinity. Comparisons of genome-wide expression between salt-sensitive and salt-tolerant plant pathogens revealed that virulence mechanisms, suboptimal in harsh environments, are commonly shut down, while essential functions such as protein quality control, osmoregulation, and iron acquisition are enhanced for survival and adaptation. Interestingly, the suppression of tomato plant diseases through salt treatment demonstrated that abiotic stress can conversely be used to control biotic stress. The current epidemiology of plant pathogens will inevitably change under future climate change scenarios. These findings provide phenotypic and genetic evidence to sufficiently understand the physiological changes in plant pathogens under abiotic stress associated with high salinity. As such, it can contribute to unraveling lethal abiotic factors against plant pathogens, thereby aiding in the development of novel defense strategies on agriculture.

## Figures and Tables

**Figure 1 plants-14-00097-f001:**
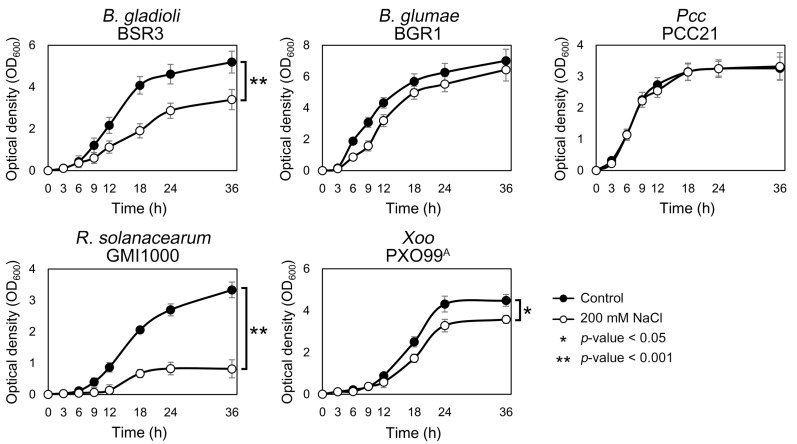
Growth curve of five plant pathogens under salt stress conditions. Bacterial cultures were grown at optimal temperatures and in media containing 200 mM NaCl. Normal medium was used as the control. Bacterial growth was monitored by measuring the optical density at 600 nm (OD_600_) using a UV spectrophotometer. Each point represents the average of three replicates, and error bars indicate the standard deviation. Asterisks (*) denote significant differences between controls and salt stress conditions.

**Figure 2 plants-14-00097-f002:**
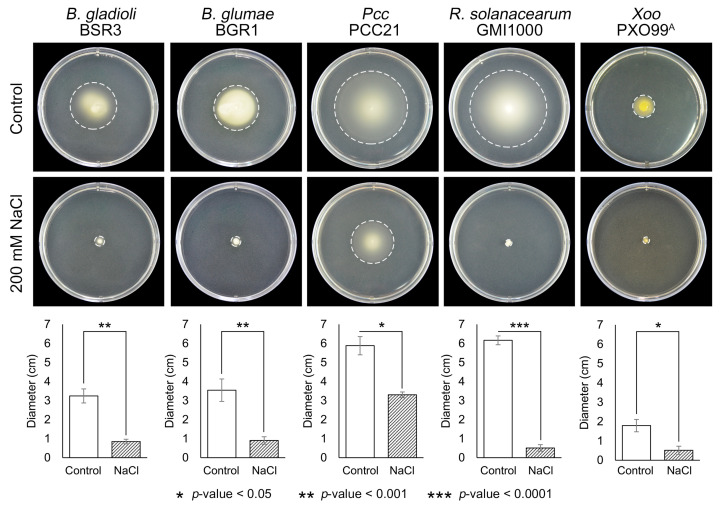
Effect of salt stress on bacterial motility. A swimming motility assay was performed to assess the bacterial motility of five plant pathogens under salt stress conditions. Cultured bacterial cells were seeded onto 0.3% agar plates containing 200 mM NaCl, and swimming motility was measured after 24 h of incubation. Uncovered plates were photographed by irradiating a standardized light source D65 (Daylight 6500 K). The bottom panel presents a graph depicting motility diameters collected from repeated experiments. Error bars denote the standard deviation, with asterisks (*) indicating statistically significant differences.

**Figure 3 plants-14-00097-f003:**
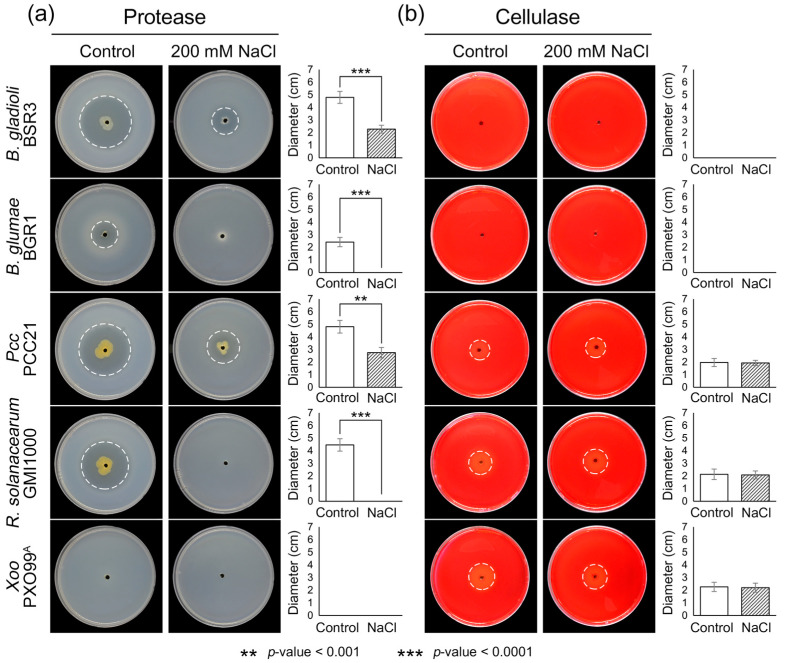
Effect of salt stress on extracellular enzyme activity. Production of extracellular enzymes in five plant pathogens was investigated under salt stress conditions. (**a**) Extracellular proteolytic activity on 1% skim milk agar medium containing 200 mM NaCl. (**b**) Extracellular cellulolytic activity on carboxymethyl cellulose agar medium containing 200 mM NaCl. Uncovered plates were photographed 3 d after inoculation with bacterial cultures. Clear zones, highlighted by white circles, indicate degradation of the substrates due to extracellular enzyme activity. The right panel presents a graph depicting activity diameters collected from repeated experiments. Error bars denote the standard deviation, with asterisks (*) indicating statistically significant differences.

**Figure 4 plants-14-00097-f004:**
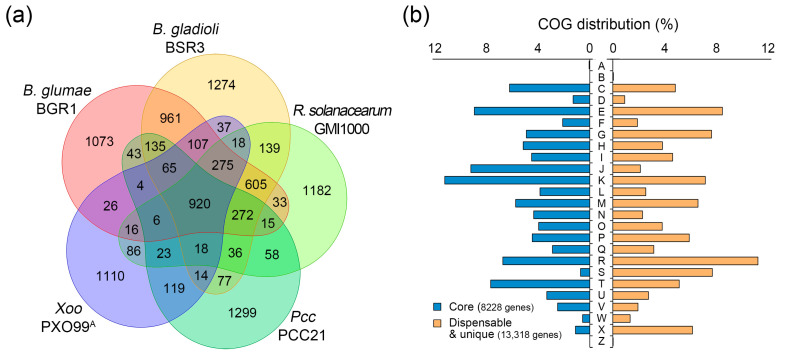
Pan-genome analysis among five plant pathogens. (**a**) Pan-genome map. Pan-genome, consisting of 26,953 genes, was analyzed using the pan-genome analysis pipeline (PGAP) tool and depicted with a Venn diagram. The innermost group represents the core genome shared by all species, whereas the outermost group represents the unique genome specific to each species. (**b**) Clusters of orthologous groups (COG) distribution between core and dispensable/unique genomes. The COG database was used for functional analysis of the pan-genome. The horizontal bar chart indicates the proportion of genes belonging to each functional category relative to the total number of genes in all COG categories. Acronyms used for COG categories: A, RNA processing and modification; B, chromatin structure and dynamics; C, energy production and conversion; D, cell cycle control, cell division, chromosome partitioning; E, amino acid transport and metabolism; F, nucleotide transport and metabolism; G, carbohydrate transport and metabolism; H, coenzyme transport and metabolism; I, lipid transport and metabolism; J, translation, ribosomal structure, and biogenesis; K, transcription; L, replication, recombination, and repair; M, cell wall/membrane/envelope biogenesis; N, cell motility; O, post-translational modification, protein turnover, chaperones; P, inorganic ion transport and metabolism; Q, secondary metabolites biosynthesis, transport, and catabolism; R, general function prediction only; S, function unknown; T, signal transduction mechanisms; U, intracellular trafficking, secretion, and vesicular transport; V, defense mechanisms; W, extracellular structures; X, mobilome: prophages, transposons; Z, cytoskeleton.

**Figure 5 plants-14-00097-f005:**
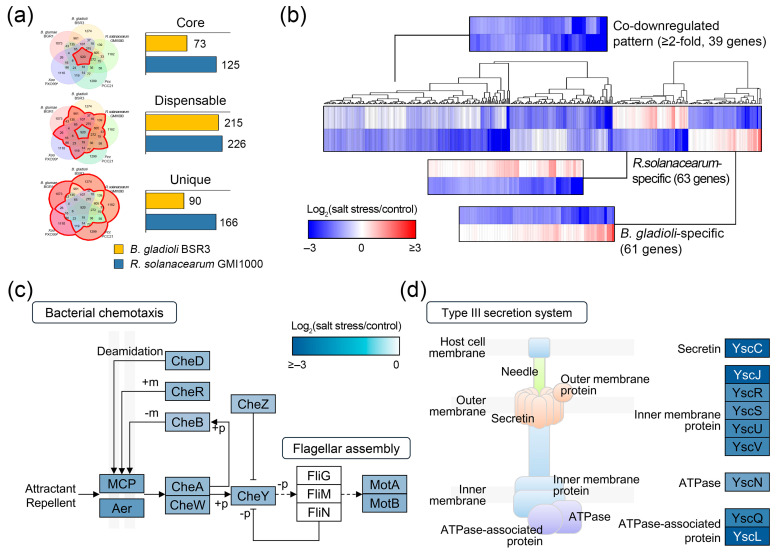
Co-downregulated patterns of salt-sensitive *B. gladioli* and *R. solanacearum*. (**a**) Distribution of downregulated differentially expressed genes (DEGs) under salt stress conditions. All DEGs of salt-sensitive *B. gladioli* and *R. solanacearum* were divided into the pan-genome map. The horizontal bar charts represent the number of downregulated DEGs distributed in the core, dispensable, and unique genomes, respectively. (**b**) Expression patterns in salt-sensitive plant pathogens. Expression of orthologous gene clusters from core and dispensable genomes, including downregulated DEGs, are shown as heatmaps with hierarchical clustering. In each cell, relative expression level is presented using a fold change value, shown as log_2_(salt stress/control). Blue and red gradients indicate decreases and increases in gene expression under salt stress conditions, respectively. The Cytoscape version 3.10.1 was used to delineate Kyoto Encyclopedia of Genes and Genomes (KEGG) systems of co-downregulated patterns, as follows: (**c**) bacterial chemotaxis, map02030; (**d**) type III secretion system, map03070. Rectangular nodes illustrate functional proteins in each system. Color differences indicate corresponding expression changes in log_2_(salt stress/control) values, with darker blue representing genes downregulated under salt stress conditions.

**Figure 6 plants-14-00097-f006:**
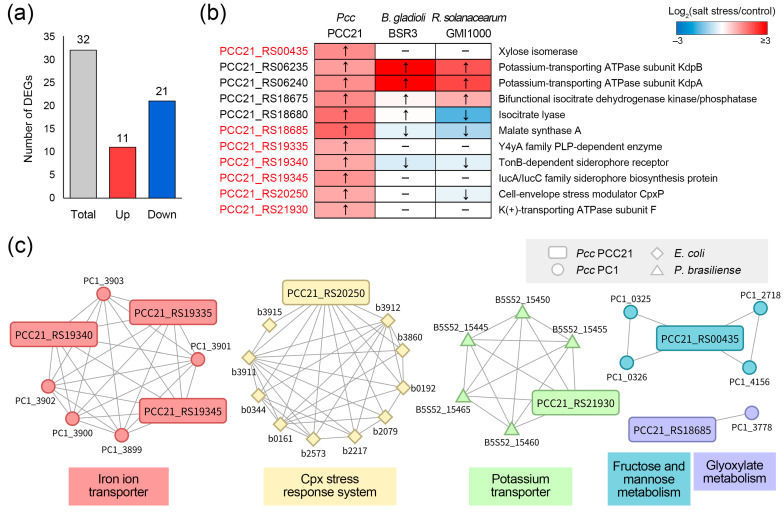
Uniquely upregulated patterns in salt-tolerant *Pcc*. (**a**) Differentially expressed genes (DEGs) of *Pcc* under salt stress conditions. The critical criteria of *p*-value < 0.05 and log_2_(salt stress/control) ≥ 1 was used for DEG identification. The bar chart illustrates the distribution of DEGs in *Pcc*. Red and blue bars denote the number of upregulated and downregulated DEGs, respectively. (**b**) Expression comparison between salt-tolerant and sensitive plant pathogens. Expression patterns of upregulated DEGs in *Pcc* were compared with those in salt-sensitive *B. gladioli* and *R. solanacearum*. The heatmap illustrates relative expression levels under salt stress conditions, shown as log_2_(salt stress/control), using a color scale showing lower (blue) to higher (red) gene expression. Upward and downward arrows in each cell indicate the upregulation and downregulation patterns under salt stress conditions, respectively. (**c**) A protein–protein interaction network for the uniquely upregulated patterns. Interactions with uniquely upregulated patterns were analyzed using the STRING database and visualized using Cytoscape version 3.10.1. The entire network contained 32 nodes and 86 edges, each obtained with a confidence score of ≥0.4. The colors of the nodes reflect the different clusters defined by Markov clustering analysis.

**Figure 7 plants-14-00097-f007:**
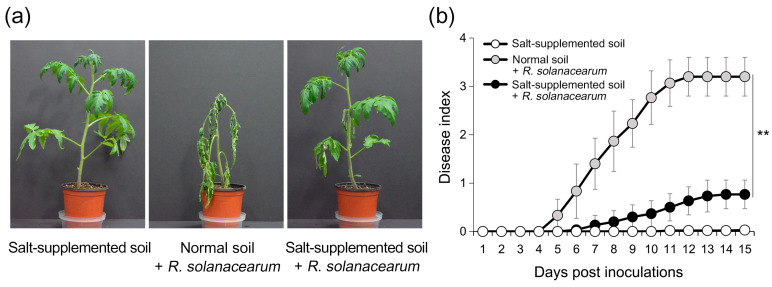
Virulence assay for *R. solanacearum* under salt stress. (**a**) Bacterial wilt disease on tomato plants. Four- to five-week-old plants grown in salt-supplemented soil were inoculated with *R. solanacearum*. Samples in salt-supplemented soil without *R. solanacearum* inoculation and samples in normal soil with *R. solanacearum* inoculation were used as controls. All samples were previously treated with the beneficial bacterium *C. salivictor* NBC122 to induce salt stress tolerance. Photographs of the disease symptoms were captured 15 d after the inoculation. (**b**) Disease severity for bacterial wilt. Daily assessment of plant disease was conducted using a disease index scale ranging from 0 (no wilting) to 4 (76–100% wilted or dead). The points on the graph represent the average disease indices of ten plants, with error bars indicating standard deviation. Asterisks (**) denote significant differences at *p*-value < 0.001 between normal and salt-supplemented soils.

**Table 1 plants-14-00097-t001:** List of plant pathogens used in this study.

Species	Strain	Tax ID ^1^	Major Host	Disease	Optimal Temperature	Source/ Reference
*Burkholderia gladioli*	BSR3	28,095	Rice	Sheath rot	28 °C	In our lab, [35]
*Burkholderia glumae*	BGR1	337	Rice	Grain rot	37 °C	In our lab, [36]
*Pectobacterium carotovorum* subsp. *carotovorum* (*Pcc*)	PCC21	555	Cabbage	Soft rot	28 °C	In our lab, [37]
*Ralstonia solanacearum*	GMI1000	305	Tomato	Bacterial wilt	28 °C	Dong-A U. of Korea, [38]
*Xanthomonas oryzae* pv. *oryzae* (*Xoo*)	PXO99^A^	64,187	Rice	Bacterial blight	28 °C	Kyung Hee U. of Korea, [39]

^1^ National Center for Biotechnology Information (NCBI) taxonomy ID.

**Table 2 plants-14-00097-t002:** Genome statistics of five plant pathogens.

Species	*B. gladioli* BSR3	*B. glumae* BGR1	*Pcc* PCC21	*R. solanacearum* GMI1000	*Xoo* PXO99^A^
Assembly	Complete	Complete	Complete	Complete	Complete
Chromosome	2	2	1	1	1
Plasmid	4	4	0	1	0
Size (bp)	9,052,299	7,284,636	4,842,771	5,810,922	5,238,555
GC (%)	67.41	67.93	52.20	66.96	63.60
Gene	8021	6623	4315	5198	4887
tRNA	69	67	76	59	54
rRNA	15	15	22	12	6
NCBI acc. ^1^	GCF_000194745.1	GCF_000022645.2	GCF_000294535.1	GCF_000009125.1	GCF_000019585.2
KEGG acc. ^1^	T01464	T00905	T02320	T00071	T00763

^1^ Genome accession number.

**Table 3 plants-14-00097-t003:** Enriched gene ontology (GO) terms for co-downregulated differentially expressed genes.

GO ID	GO Term	Count	*p*-Value
GO:0042128	Nitrate assimilation	8	3.58 × 10^−8^
GO:0034053	Modulation by symbiont of host defense-related programmed cell death	5	4.26 × 10^−7^
GO:0006935	Chemotaxis	15	5.51 × 10^−7^
GO:0007165	Signal transduction	12	1.27 × 10^−5^
GO:0071249	Cellular response to nitrate	3	0.0005
GO:0051603	Proteolysis involved in cellular protein catabolic process	4	0.0008
GO:0015807	L-Amino acid transport	4	0.0015
GO:0019646	Aerobic electron transport chain	5	0.0035
GO:0006572	Tyrosine catabolic process	3	0.0038
GO:0009306	Protein secretion	6	0.0051
GO:0043041	Amino acid activation for non-ribosomal peptide biosynthetic process	4	0.0059
GO:0032981	Mitochondrial respiratory chain complex I assembly	4	0.0085
GO:0030254	Protein secretion by the type III secretion system	3	0.0091

## Data Availability

The original RNA-seq data presented in this study are openly available in NCBI Gene Expression Omnibus (GEO) database (https://www.ncbi.nlm.nih.gov/geo/, 24 July 2024) under accession number GSE272984. Genomic sequences of plant pathogens were downloaded from the NCBI RefSeq database (https://www.ncbi.nlm.nih.gov/refseq/, 15 June 2023) under accession numbers: GCF_000009125.1, GCF_000019585.2, GCF_000022645.2, GCF_000194745.1, and GCF_000294535.1).

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
