# Peer review of "Understanding the Impact of Salt Stress on Plant Pathogens Through Phenotypic and Transcriptomic Analysis"

_plants, 2025, doi:10.3390/plants14010097_

Round 1
Reviewer 1 Report
Comments and Suggestions for Authors
This study investigates the effects of salt stress on representatives of some plant pathogens. The study also concludes that pan-genome-based comparative transcriptomics have elucidated the genetic interpretation of different responses among plant pathogens. Overall, the research and experiments are clearly described and the results of the research are interesting. However, there are some minor flaws in the manuscript listed below:
-According to line 327, a total of 18 RNA-seq libraries were constructed for sequencing the transcriptomes of salt-sensitive (B. gladioli and R. solanacearum) and salt-tolerant (Pcc) plant
Pathogens. What is the contribution of each pathogen for RNA-seq libraries? How many biological or technical replications are considered?
- I suggest rewriting the conclusions to reflect the results and emphasizing the practical applications of these findings in agriculture and environmental changes.
Author Response
This study investigates the effects of salt stress on representatives of some plant pathogens. The study also concludes that pan-genome-based comparative transcriptomics have elucidated the genetic interpretation of different responses among plant pathogens. Overall, the research and experiments are clearly described and the results of the research are interesting. However, there are some minor flaws in the manuscript listed below:
>>> We thank the reviewers for valuable comments and suggestions for improvement. Below, we address point by point the major issue raised, and indicate the changes made in the accompanying revised manuscript. All changes in the revised manuscript are marked in red.
Comments 1: According to line 327, a total of 18 RNA-seq libraries were constructed for sequencing the transcriptomes of salt-sensitive (B. gladioli and R. solanacearum) and salt-tolerant (Pcc) plant pathogens. What is the contribution of each pathogen for RNA-seq libraries? How many biological or technical replications are considered?
Response 1: Thank you for your valuable feedback. For each plant pathogen, we constructed RNA-seq libraries with three biological replicates in both the control (0 mM NaCl) and treatment (200 mM NaCl) conditions. The RNA-seq libraries for B. gladioli, R. solanacearum, and Pcc contained an average of 45,447,416, 38,559,098, and 66,648,169 paired-end reads, respectively. Based on the reviewers' comments, we have addressed additional experimental details for the RNA-seq libraries and included statistical results specific to each plant pathogen’s libraries, rather than the entire dataset, in the section “2.5. Co-downregulated patterns in salt-sensitive plant pathogens”, lines 332–337.
Comments 2: I suggest rewriting the conclusions to reflect the results and emphasizing the practical applications of these findings in agriculture and environmental changes.
Response 2: Thank you for pointing this out. Based on the reviewer's comments, we have revised the entire Conclusion section. Instead of repeating the results, we have emphasized the importance and implications of the results for agriculture in the context of environmental change. The current epidemiology of plant pathogens will inevitably change under future climate change scenarios. In this regard, we believe that these findings will provide valuable information for understanding and exploiting key physiological changes in plant pathogens under abiotic stresses such as high salinity. These revisions can be found in section “4. Conclusions”, lines 679–695.
Reviewer 2 Report
Comments and Suggestions for Authors
1. Line 81-94:
The aim of study is unclear and need to rewrite in briefly and just focus what are you aimed from this study.
2. The section: “2.2.Effects of salt stress on bacterial motility”:
Rewrite this section by mentioning to your results then you can discuss the results
3. The section: “2.3. Effects of salt stress on extracellular enzyme production”:
Rewrite this section by mentioning to your results then you can discuss the results
4. The section: "3.1. Plant pathogens and culture conditions":
The author must mention to resource of plant pathogens in table 1, that used here
5. This section:”3.9. Virulence assay under salt stress conditions”:
You are mentioned in first for several plant bacterial pathogens , why are you just estimated one plant bacterial pathogen under salt stress in this section
6. The conclusions are poorly and unclear , therefore, need to rewrite in briefly without repeat results
Author Response
We thank the reviewers for valuable comments and suggestions for improvement. Below, we address point by point the major issue raised, and indicate the changes made in the accompanying revised manuscript. All changes in the revised manuscript are marked in red.
Comments 1: (Line 81-94) The aim of study is unclear and need to rewrite in briefly and just focus what are you aimed from this study.
Response 1: We sincerely appreciate the reviewer’s valuable comments and suggestions, which have greatly contributed to improving the quality and clarity of the manuscript. Based on the reviewer's comments, we have revised the final paragraph of the “Introduction” to concisely describe the methods and results of the study and to clearly state the study’s purpose. The purpose of this study was to elucidate the diverse interactions of representative plant pathogens in response to salt stress and to further investigate the biological mechanisms regulated by salt stress through cross-species comparative transcriptome analysis. We believe these findings provide important insights for agricultural environments under climate change scenarios and hope that this work will be suitable for publication in the journal “Plants”. These revisions can be found in section “1. Introduction”, paragraph 5, lines 81–91.
Comments 2: (The section: “2.2. Effects of salt stress on bacterial motility”) Rewrite this section by mentioning to your results then you can discuss the results.
Response 2: Thank you for your valuable feedback. Following the reviewer’s comment, we have carefully reviewed and revised section “2.2. Effects of salt stress on bacterial motility” by restructuring the content to present the results first, followed by a detailed discussion of the findings. To ensure clarity, we have removed unnecessary sentences and enhanced the logical flow. This restructuring enhances the coherence of the section, making the results and discussion more straightforward and easier to follow. These revisions can be found in section “2.2. Effects of salt stress on bacterial motility”, lines 165–204.
Comments 3: (The section: “2.3. Effects of salt stress on extracellular enzyme production”) Rewrite this section by mentioning to your results then you can discuss the results.
Response 3: Thank you for your valuable feedback. Following the reviewer’s comment, we have carefully reviewed and revised section “2.3. Effects of salt stress on extracellular enzyme production” by restructuring the content to present the results first, followed by a detailed discussion of the findings. To ensure clarity, we have removed unnecessary sentences and enhanced the logical flow. This restructuring improves the coherence of the section, making the results and discussion more straightforward and easier to follow. These revisions can be found in section “2.3. Effects of salt stress on extracellular enzyme production”, lines 206–234.
Comments 4: (The section: "3.1. Plant pathogens and culture conditions") The author must mention to resource of plant pathogens in table 1, that used here.
Response 4: We appreciate the reviewer’s valuable suggestions. Based on your recommendation, we have added the bacterial source information for five representative plant pathogens used in this study to Table 1. Specifically, three plant pathogens—Burkholderia gladioli BSR3, Burkholderia glumae BGR1, and Pectobacterium carotovorum subsp. carotovorum PCC21—are part of our laboratory’s collection. In contrast, Ralstonia solanacearum GMI1000 and Xanthomonas oryzae pv. oryzae (Xoo) PXO99A were previously obtained from Dong-A University and Kyung Hee University in the Republic of Korea, respectively.
It should be noted that R. solanacearum and Xoo were not obtained specifically for this study but were previously acquired for other research projects. Xoo was used in a study investigating the functional organization of the type 6 secretion system [1], while R. solanacearum was used to explore global virulence factors through cross-species comparative in planta transcriptomics [2]. These prior studies have laid a strong foundation for the current work. These revisions can be found in the sections “Table 1”, line 124 and “3.1. Plant pathogens and culture conditions”, lines 541–544.
Related references:
- Choi, Y.; Kim, N.; Mannaa, M.; Kim, H.; Park, J.; Jung, H.; Han, G.; Lee, H.H.; Seo, Y.S. Characterization of type VI secretion system in Xanthomonas oryzae pv. oryzae and its role in virulence to rice. Plant Pathol J 2020, 36, 296, doi:10.5423/PPJ.NT.02.2020.0026.
- Park, J.; Jung, H.; Mannaa, M.; Lee, S.Y.; Lee, H.H.; Kim, N.; Han, G.; Park, D.S.; Lee, S.W.; Lee, S.W.; et al. Genome-guided comparative in planta transcriptome analyses for identifying cross-species common virulence factors in bacterial phytopathogens. Front Plant Sci 2022, 13, 1030720, doi:10.3389/FPLS.2022.1030720.
Comments 5: (This section:”3.9. Virulence assay under salt stress conditions”) You are mentioned in first for several plant bacterial pathogens, why are you just estimated one plant bacterial pathogen under salt stress in this section.
Response 5: Thank you for bringing this to our attention. The first goal of this study was to elucidate the fundamental interactions between plant pathogens and salt stress through cross-species phenotypic and transcriptome analyses. Additionally, we aimed to explore the potential of using salt stress as a means to control plant pathogens. However, it is important to note that salt treatment affects not only plant pathogens but also plant hosts, and further research is needed to establish effective co-treatments with beneficial bacteria to protect the host plants.
For this reason, we selected R. solanacearum, which is the most sensitive to salt stress among five representative plant pathogens, in the virulence assay. Under salt stress conditions, R. solanacearum exhibited a drastic decrease in growth, with many genes related to its virulence systems being downregulated. As expected, we confirmed that salt treatment in soil was effective in controlling bacterial wilt diseases caused by R. solanacearum. These findings provide valuable insights that can be used to develop new strategies for managing plant diseases. Based on the reviewer's comments, we have clarified the rationale for selecting R. solanacearum in the virulence assay in sections "2.7. Effect of salt stress on the virulence of R. solanacearum", lines 506–510 and "3.9. Virulence assay under salt stress conditions", lines 651–653.
Comments 6: The conclusions are poorly and unclear, therefore, need to rewrite in briefly without repeat results.
Response 6: Thank you for pointing this out. Based on the reviewer's comments, we have revised the entire Conclusion section. Instead of repeating the results, we have emphasized the importance and implications of the results for agriculture in the context of environmental change. The current epidemiology of plant pathogens will inevitably change under future climate change scenarios. In this regard, we believe that these findings will provide valuable information for understanding and exploiting key physiological changes in plant pathogens under abiotic stresses such as high salinity. These revisions can be found in section “4. Conclusions”, lines 679–695.
Reviewer 3 Report
Comments and Suggestions for Authors
The authors aim to investigate the effects of salt stress on the phenotypic and transcriptomic responses of several plant pathogens. Revealing that while some pathogens like Burkholderia gladioli and Ralstonia solanacearum are sensitive to salt stress, others like Pectobacterium carotovorum subsp. carotovorum exhibit tolerance. That may have potential implications for managing plant diseases under climate change scenarios.
The work enclosed in MS is quite wide-ranging and considers several pathogen species. I missed some information, or it was not emphasized enough in the multitude of others: Why was this particular salt concentration chosen for testing? How often can it be found in soils? And what is the occurrence of such salt-supplemented soils in general?
In Figure 7, please add for comparison a photo of a plant at the same developmental stage, but not infected with the pathogen. Plants grown on salt-supplemented soil will be enough.
Overall, the presented research results are worth publishing and I think they will be of interest to Plants readers.
Author Response
The authors aim to investigate the effects of salt stress on the phenotypic and transcriptomic responses of several plant pathogens. Revealing that while some pathogens like Burkholderia gladioli and Ralstonia solanacearum are sensitive to salt stress, others like Pectobacterium carotovorum subsp. carotovorum exhibit tolerance. That may have potential implications for managing plant diseases under climate change scenarios.
>>> We thank the reviewers for valuable comments and suggestions for improvement. Below, we address point by point the major issue raised, and indicate the changes made in the accompanying revised manuscript. All changes in the revised manuscript are marked in red.
Comments 1: The work enclosed in MS is quite wide-ranging and considers several pathogen species. I missed some information, or it was not emphasized enough in the multitude of others: Why was this particular salt concentration chosen for testing? How often can it be found in soils? And what is the occurrence of such salt-supplemented soils in general?
Response 1: Thank you for pointing this out. As climate change intensifies worldwide, many studies are trying to understand the various impacts of abiotic stresses such as salt stress on agricultural environments. Among abiotic stresses, soil salinity can be further aggravated by both natural and anthropogenic factors [1]. Climate change along with global warming increases natural salinization processes, and agricultural activities, especially irrigation systems, accelerate these processes. In this regard, this study aims to address the serious challenges posed by extreme salt stress conditions rather than focusing on the current salinity levels in agricultural environments.
In particular, high soil salinity at 200 mM NaCl has been shown to significantly affect the growth, development, and yield of various plants, including tomato and wheat [2,3]. In microorganisms, the concentration of 200 mM NaCl has also been used as a benchmark in studies exploring the effects of salt stress [4,5]. Therefore, we selected 200 mM NaCl as the standard concentration in this study, as it is widely applied in salt stress research. We believe that these findings are essential for understanding global physiological changes in plant pathogens under salt stress, which may contribute to the development of strategies to mitigate plant diseases under future climate change scenarios. We address these issues in the section “2.1. Salt stress tolerance of representative plant pathogens”, lines 95–103 and 126–133.
Related references:
- Stavi, I.; Thevs, N.; Priori, S. Soil salinity and sodicity in drylands: A review of causes, effects, monitoring, and restoration measures. Front Environ Sci 2021, 9, 712831, doi:10.3389/FENVS.2021.712831/BIBTEX.
- Orhan, F. Alleviation of salt stress by halotolerant and halophilic plant growth-promoting bacteria in wheat (Triticum aestivum). Brazilian Journal of Microbiology 2016, 47, 627, doi:10.1016/J.BJM.2016.04.001.
- Habibi, N.; Terada, N.; Sanada, A.; Koshio, K. Alleviating salt stress in tomatoes through seed priming with polyethylene glycol and sodium chloride combination. Stresses 2024, 4, 210–224, doi:10.3390/STRESSES4020012.
- Naamala, J.; Subramanian, S.; Msimbira, L.A.; Smith, D.L. Effect of NaCl stress on exoproteome profiles of Bacillus amyloliquefaciens EB2003A and Lactobacillus helveticus EL2006H. Front Microbiol 2023, 14, 1206152, doi:10.3389/FMICB.2023.1206152.
- Philips, J.; Rabaey, K.; Lovley, D.R.; Vargas, M. Biofilm formation by Clostridium ljungdahlii is induced by sodium chloride stress: Experimental evaluation and transcriptome analysis. PLoS One 2017, 12, e0170406, doi:10.1371/JOURNAL.PONE.0170406.
Comments 2: In Figure 7, please add for comparison a photo of a plant at the same developmental stage, but not infected with the pathogen. Plants grown on salt-supplemented soil will be enough.
Response 2: Thank you for your valuable feedback. We agree with this comment. We have already measured the growth and development of uninoculated tomato plants grown in salt-supplemented soil at the same developmental stage. Uninoculated tomato plants showed normal growth and development due to the protective effect of salt stress by the plant growth-promoting C. salivictor. These data clearly show that soil salinity significantly reduces the virulence of R. solanacearum, thereby significantly resulting in a notable decrease in disease symptoms in tomato plants. Following the reviewer’s comments, we have reorganized the results for uninoculated tomato plants into the main figure and revised the relevant content throughout the manuscript (sections “Figure 7 legend”, lines 531–532 and “3.9. Virulence assay under salt stress conditions”, line 666).
Round 2
Reviewer 2 Report
Comments and Suggestions for Authors
NA